# Influence of the Coarse Grain Structure of a Titanium Alloy Ti-4Al-3V Formed by Wire-Feed Electron Beam Additive Manufacturing on Strain Inhomogeneities and Fracture

**DOI:** 10.3390/ma16113901

**Published:** 2023-05-23

**Authors:** Vasily Klimenov, Evgeny Kolubaev, Klopotov Anatoly, Andrey Chumaevskii, Artem Ustinov, Irina Strelkova, Valery Rubtsov, Denis Gurianov, Zeli Han, Sergey Nikonov, Andrey Batranin, Margarita Khimich

**Affiliations:** 1National Research Tomsk Polytechnic University, Division for Materials Science at SAMT TPU, Lenina pr. 30, Tomsk 634050, Russia; klimenov@tpu.ru (V.K.); strelkova@tpu.ru (I.S.);; 2Institute of Strength Physics and Materials Science Siberian Branch of Russian Academy of Sciences, Akademicheskiy pr. 2/4, Tomsk 634055, Russia; eak@ispms.ru (E.K.);; 3Tomsk State University of Architecture and Building, Department of Applied Mechanics and Materials Science, Solyanaya Sq. 2, Tomsk 634003, Russia

**Keywords:** additive manufacturing, electron beam, titanium alloy, macrostructure, microstructure, mechanical properties, tomography, non-destructive testing

## Abstract

In this work, based on the multilevel approach, the features of the structure and properties of titanium alloy, formed during high-performance additive manufacturing by wire-feed electron beam technology, were studied. Methods of non-destructive X-ray control and tomography, along with optical and scanning electron microscopy, were used to study the structure at different scale levels of the sample material. The mechanical properties of the material under stress were revealed via the simultaneous observation of the peculiarities of deformation development, using a Vic 3D laser scanning unit. Using microstructural and macrostructural data, as well as fractography, the interrelations of structure and material properties caused by the technological features of the printing process and the composition of used welding wire were revealed.

## 1. Introduction

Titanium, due to its good combination of physical, mechanical, and biomedical properties, is the most suitable material for a wide range of applications in medicine as implants, endoprostheses, plates for internal fixation, and cranioplasty, as well as for various fixation elements (bridges, braces, screws, etc.) in traumatology and dentistry [1]. In order to expand the use of titanium alloys in medicine and to meet the ever-increasing demands for them and for the associated products, new titanium alloys are being created; the properties of existing alloys are also being improved by applying coatings to them [2,3], as well as by the use of additive technologies with unique shaping and structuring properties [4,5,6]. Currently, additive technologies or 3D printing using titanium alloys are increasingly being employed in various industries and in medicine, raising new issues regarding the efficiency and reliability of products, especially in extreme conditions and with complex influencing factors. This situation requires an understanding of the processability of the formed material, which usually acquires a specific structure and properties different from those obtained using traditional methods [7,8,9,10,11]. It is known that the characteristic values of cooling rates for the majority of traditional methods of parts production can be in the range of 48–4∙10^−3^ K/s, at which point significant changes in the structure and properties of the formed material occur [7]. In the case of production using additive technologies, the melt-cooling rates can be in the range of 10^3^–10^8^ K/s; in addition, the temperature gradient can reach values of 10^6^ K/cm [10]. It should also be noted that thermal cycling conditions, which are characteristic of additive technologies, are caused by the layer-by-layer alloying of metals and alloys. All these factors together cause the complexity of the processes accompanying the formation of the composition, affecting the structure and properties of both the forming material and the product as a whole. Therefore, the issues of controlling the composition and structure of metals and alloys at various scales [11], including defects [12,13,14], appear to be important and necessary, both for predicting properties and for the certification of the material or technology as a whole, as well as for compliance with national standards [15]. The efficiency of the applied 3D printing technology for titanium alloys is also important. One such application is the use of metal wires for printing [12,13,15,16,17,18,19,20,21]. At the same time, special attention should be paid to the use of titanium alloys for this purpose in the form of welding wire, which makes the process more accessible, and, to some extent, cheaper [16,22]. Among the two most widely used methods—laser and electron beam techniques—the latter is becoming more and more widespread, especially in relation to the application of titanium alloys and stainless steels. This is mainly due to the fact that under the conditions of electron beam exposure, which can mostly be carried out in a vacuum, the most suitable materials are easily oxidizable alloys, such as those mentioned above. Due to the widely studied features of the structure and the stress state in metal alloys, including titanium, which are formed under conditions of high-energy laser or electron beam irradiation, there is already a clear understanding of the optimal values of the applied radiation power, its scanning speed, and the size of the layer thickness and layer overlap during the formation of a bulk object [23], meaning that it is possible to obtain products with satisfactory properties [10,11,12,13,14,15,16,17,18,19,20,21,24,25]. At the same time, the strength properties and hardness of the formed materials often exceed those of alloys obtained by casting and rolling [23,26]. The development of printing strategies, in terms of printing small objects, and the monitoring of melt baths for parts with complex shapes make it possible to obtain objects with controlled porosity [10] and parts with a microstructure [24], making them acceptable for use in the manufacture of critical parts. At the same time, the possibility of improving the structure and ductility of titanium alloys, formed under the conditions of additive technologies, by means of subsequent heat or thermomechanical treatment is proposed as the most suitable solution [15,26,27]. However, using the most powerful additive technologies available, based on the use of wire, such treatments will significantly affect the efficiency of manufacturing large-sized products, which, in addition, is difficult and necessitates qualitative heat treatment, especially in the case of parts with a complex shape and geometry. Recently, there have been works published that focus on the formation of an initially effective structure in titanium alloys, providing the sufficiently high tensile strength characteristics of commercial titanium alloy through special alloying [28], or by means of minor heat treatment [29]; we believe that these conditions can be realized in the course of printing. The excellent properties of the resulting materials are due to the unusual formation of specific micro- and nanostructures that are rarely observed in traditionally treated titanium alloys. The aim of this work is to review the processes of high-performance electron beam printing with welding wire, along with the evaluation of defects, microstructure, and the properties of titanium alloy, including fractures. The objective set in the work is achieved on the basis of a comprehensive study of the composition, structure, defects, and strength properties of the titanium alloy Ti-4Al-3V formed under electron beam exposure, with an assessment of the tensile strain characteristics using the Vic-3D system and fractures, based on the study of fracture surface fractography.

## 2. Materials and Methods

Titanium alloy Ti-4Al-3V (Ti-6Al-4V alloy welding wire) in the form of 1.6 mm filament was selected as the material for sample printing; its composition is presented in Table 1, in comparison with other titanium alloys that are most commonly used in additive technologies.

It should be noted that the elemental composition of the wire meets the requirements of ISO 24034:2020, but the contents of the main alloying elements influencing the hardening of the titanium alloy correspond to the lowest values.

The samples were obtained in the laboratory electron beam additive manufacturing setup developed at ISPMS SB RAS, via the process of the layer-by-layer fusion of wire on a substrate in the form of a 150 × 60 × 2.5 mm plate of Grade 2 material (Figure 1a). The substrate was placed on a protective stainless steel sheet (160 × 60 × 5 mm) and everything was fixed to a cooled three-coordinate work table with the help of metal clamps. The table was liquid-cooled and the temperature of the table was kept at 13–15 °C during the printing process. The process of pattern formation was carried out in a vacuum at a pressure of 10^−3^–10^−2^ Pa. To form each layer, an electron beam and source material—a wire of 1.6 mm diameter, made of Ti-4Al-3V alloy (Figure 1a)—were sent to the printing zone. The wire and the previous layer (the substrate when printing the first layer) were melted by the electron beam, forming a melt bath. The sample was formed by moving the printing area and the work table along a given trajectory. During the production of this sample, layers were applied alternately in opposite directions, according to the scanning scheme shown in Figure 1b. The overall view of the printed sample, which is in the form of a wall, is shown in Figure 1c. The appearance of the sample clearly shows two characteristic areas: the area of directed beam movement (Figure 1c-8) and the area of beam reversal (Figure 1c-9).

The parameters of the electron beam, printing speed, and wire feed rate were selected, based on the parameters of the material, substrate, sample size, and thickness of the formed layer to ensure a stable printing process. The decrease in electron beam current with increasing sample height was caused by changing the conditions of heat dissipation from the melt bath. The maximum current value was used when printing the first layer, to heat the substrate and form a melt bath within it. In addition, the beam current decreased exponentially from 55 to 40 mA as it moved away from the substrate to avoid overheating the material (Figure 2). The printing parameters are shown in Table 2. The beam characteristics and printing modes corresponded to the optimal values obtained in previous studies and ensured the formation of a porosity-free material [19].

Blades for the tensile tests and examinations (Figure 3a,b), along with rectangular prisms for the construction of 3D metallographic models of different sections of the examined samples and for micro-focus tomography (Figure 3c), were fabricated from the printed samples to study the defects, the structural phase state of the printed sample material, and its physical-mechanical properties.

Nondestructive control of the printed plates was carried out with the help of a portable pulsed X-ray device (RAP 160 No. 145 (Diagnostika-M, Tomsk, Russia)) and a flat panel digital detector GE DXR250C-W (GE, Boston, MA, USA) according to the control method in accordance with ISO 17636-2-2017. The control parameters are shown in Table 3.

Nondestructive control of the wire and printed samples for metallography and mechanical testing and a tomographic examination for their micro- and macro-defects was carried out on a computer X-ray “Orel-MT” microtomograph (University TPU, Tomsk, Russia). This unit is equipped with an XWT 160-TC X-ray tube (X-ray WorX, Garbsen, Germany) and a PaxScan-2520V detector panel (Varian medical system, Palo Alto, CA, USA) with a positioning control system. The samples were scanned according to the following parameters: acceleration voltage—130 kV, current—27 μA, resolution—11.3 μm, number of projections—1200 units, step size—0.3, and copper filter—2 mm. Tomographic reconstruction was performed using the Nrecon software product developed by Bruker for micro-CT. After reconstruction, the tomograms were segmented to obtain two models, representing the sample material itself and the internal porosity [30].

The preparation of samples for metallographic studies and X-ray structural analysis was conducted by cutting them out of various sections of the printed plate (Figure 3) and subsequently grinding the surfaces, using abrasive paper with a consistently decreasing abrasive grain size. Final polishing was performed on a cloth, using an aqueous suspension of either chromium oxide or diamond paste.

The X-ray diffraction analysis was performed with an XRD-6000 (Shimadzu, Kioto, Japan) diffractometer using Cu-*K*_α_ radiation. The voltage that was applied to the tube was 40 kV and the current was 30 mA. The scan range of 2θ angles was 10–100°, the scan step was 0.05°, and the exposure time was 1 s. The obtained X-ray diffraction patterns were processed using the Rietveld refinement. Calculations were performed for the lattice parameters, unit cell volumes, the size of coherent scattering regions (CSR), and microstresses (type II stresses and internal grains).

A Thermo Scientific Niton XL3t GOLDD + XRF analyzer was used to conduct an elemental analysis of the initial (wire) and AT-produced materials.

Etching of the samples for metallographic studies after polishing was performed using a Kroll reagent consisting of 10 mL HNO_3_, 3 mL HF, and 87 mL distilled water. The microstructure of the samples was analyzed using an optical microscope Axio Observer A1m (Zeiss, Oberkochen, Germany). Hardness measurement was carried out using the Duramin-5 device (Struers, Ballerup, Denmark) at a load of 100 g, with an action time of 12 s.

The tensile tests on the specimens were performed on the INSTRON 3386 testing machine (INSTRON Corp., Glenview, IL, USA). The evolution of deformation fields during the tests was recorded using the VIC-3D optical measurement system (Correlated Solutions, Irmo, SC, USA). The application of the digital image correlation method in the study of deformation field characteristics is a very promising approach by which to elucidate the peculiarities of the deformation processes of metals and alloys with different structural phase states [29].

The displacement fields recorded by the VIC-3D optical measurement system reflect the projections of the displacements of local surface areas along the OX axis (“transverse deformation” of the specimens) and along the OY axis (“longitudinal deformation” of the specimens) (Figure 4).

## 3. Results and Discussions

Radiography of the printed plates using a portable pulsed X-ray RAP 160 with a resolution of 200 µm did not reveal any defects (Figure 5). The heterogeneity of the color image is associated with the different thicknesses of the sample (bright areas correspond to thicker zones), which is clearly confirmed by the fact that the sample is thinner in the lower part, where the melt hardens faster. This type of heterogeneity in the surface structure is also clearly visible from the visual inspection of the printed samples. Radiography of the specimens prepared for mechanical testing, which was carried out with the help of the “Orel-MT” microtomograph with a sensitivity of up to 15 microns, allowed us to reveal the presence of a small number of pores with sizes of less than 200 microns on some specimens (Figure 6). In comparison with the defectiveness of the material obtained in additive technology (AT) conditions with the use of powder materials [30], it should be noted that the material is of higher quality, both in terms of total defectiveness and in terms of the absence of defects due to the specifics of the starting material.

The computed tomography of the original wire and the printed material samples also showed the high quality of the original and printed material (Figure 7). It should be noted that this type of research in AT is very important as it allows us to approach the application of machine learning in the quality control of materials and products in general.

For microstructural studies, it is very important to control the elemental composition of both the starting material and the material formed under the influence of the electron beam, the power density of which can significantly affect the material composition. Analysis of the elemental composition of the Ti-4Al-3V wire material showed a significant reduction in both Al and V in the titanium alloy (Table 4).

Efficiently selected printing modes resulted in minimal changes to the elemental composition of the clad material. X-ray diffraction and X-ray fluorescence analyses allowed us to determine the elemental and phase composition of the obtained material (Figure 8). There is a basic α-Ti phase with the insignificant presence of a β-Ti phase. At the same time, the levels of concentration of the alloying elements in the starting material did not allow for obtaining hardness characteristics corresponding to the values obtained with Ti-6Al-4V alloys. It should be noted that the metallographic analysis also showed a difference in the microstructure of the printed alloy from the wire and that previously obtained from alloys such as Ti-6Al-4V.

In the diffractogram of the sample Ti-4Al-3V taken from the lower surface, it can be seen that the phase composition is represented by the main phase, α-Ti, where there is a “trace” of β-Ti.

As can be seen, there are noticeable differences between the starting material—wire—and the different regions of the grown sample. First of all, it should be noted that the wire contains traces of the β-phase in addition to the main phase, as well as an unidentified reflection of the impurity phase (Table 5). In samples cut from the side surface that were oriented horizontally, the impurity phases are absent; only the main phase, α-Ti, and small amounts of β-phase can be observed. A significant redistribution of the intensities of the main α-Ti phase can be observed, depending on the section of the studied object from which the samples were cut for study. The diffractogram obtained from the wire clearly shows a predominant orientation in the <001> direction—the direction of wire formation characteristic of the hexagonal crystal lattice. In the lower part of the sample, a pronounced predominant orientation in the same <001> direction can be seen—the texture is in a direction perpendicular to the substrate and parallel to the growth vector of the product. In the upper part of the sample, the preferential orientation of the planes (002) is preserved, but the texture is not as pronounced as in the lower part. This may be due to the changing conditions of heat dissipation during growth—the lower part of the sample is periodically heated and cooled during the deposition of the overlying layers, but the heating temperature decreases as the number of layers increases. In the upper part of the sample, the number of such cycles is less than in the lower part. Thus, the repeated heating and cooling in both parts of the object contribute to the growth of the grain structure of the basic α-phase in the growth direction of the sample; however, in the lower part, the volumetric growth of the α-phase is reduced (Table 6 and Table 7). Samples cut from the side surface of the sample do not show a similar pattern. The heat exchange conditions there are different—in addition to repeated heating and cooling, the temperature of which changes with each new layer applied, the side surface is also the interface between the solid and gaseous phases, an area where the heat exchange conditions change significantly. In addition, since the application of each new layer is accompanied by a local transition of the wire to the liquid phase, in some areas of the lateral surface, there is a boundary with three phases, namely, solid/liquid/gas. Such temperature conditions of product formation lead to a change in the crystallographic orientations of the growing crystallites, which, in turn, leads to the redistribution of intensities of the main phase reflexes on diffractograms. In samples cut according to the horizontal plane of the studied object, this preferential orientation is observed in the plane that is parallel to the substrate. The most pronounced plane family is (100). At the same time, the intensity of plane (002) in such samples is minimal. Such intensity behavior indicates that the thickness of the examined specimens does not exceed the thickness of an applied layer.

To confirm the above reasoning, texture coefficients were calculated for each sample that was examined. Planes (100), (002), and (101) of the α-Ti phase were chosen as the most representative and informative. Figure 9 shows how the texture coefficients of these planes will change, depending on the subject.

As can be seen, the most predominant orientation in the <001> direction is characteristic of the wire, which is associated with the method of its production. A similar texture is most pronounced in the lower part of the object and is practically not observable in the horizontal plane in comparison with the rest of the samples examined. The lateral surface, as mentioned above, is characterized by a marked redistribution of intensities, while family (100) is the most strongly oriented in the horizontal plane.

The microstructure of the most characteristic areas of the printed material is shown in Figure 10 and Figure 11. The microstructure of the sample that was cut from the edge of the printed wall consists of columnar grains of a diameter d = 0.8–2.0 mm, oriented in the opposite direction to the heatsink parallel to the *z*-axis. Each columnar grain has alternating black layers at a distance of h = 0.8–2.0 mm, with a crushed structure formed by the passage of the electron beam and the formation of each new layer. The light layers represent a Widmannstett non-equilibrium basket-type structure of α′ phase colonies with different crystallographic orientations, exhibiting a high density of dislocations and twins. The formation of such a structure is indicative of the overheating of the liquid melt with temperatures in the β-area of above 1020 °C, along with the subsequent accelerated cooling due to the additional influence of the surrounding atmosphere at the edge of the sample wall. Such a structure has inflated values of hardness (375–380 HV) and strength.

The microstructure and properties differ from the cast edge of the sample wall as it moves away from the edge of the wall, closer to the center. The microstructure becomes a mixture of small plates of the α′-phase and lamellae of the (α + β) phase plates with increased hardness (363 HV). With a greater distance from the cast edge of the wall, decomposition of the α′-phase occurs, which indicates the duration for holding the cooled metal at melting temperatures in the region of the (α + β) phase, corresponding to a temperature of 950 °C. The formation of the β-phase leads to a decrease in hardness (223–324 HV) and strength.

Tensile specimens cut from the central part of the printed wall have a columnar grain structure growing vertically through several layers (Figure 12, Figure 13 and Figure 14). The lamellar microstructure of the columnar grains consists of colonies of (α + β) phases of different lengths and widths; the lamellae of the β-phase are smaller in size and are located between the plates of the α-phase. The formation of the lamellar structure indicates prolonged exposure to temperatures in the region of the (α + β) phase, as a result of the superposition of subsequent metal layers of molten wire and due to maintaining a constant temperature of 900 °C.

The hardness values are not uniformly distributed across the width of the columnar structure. It is possible to assume that differences in microhardness depend on the size of the plates of phases (α + β), and also that they are a consequence of a change in the crystal-graphic orientation of the colonies of plates. Near the edges of the columns, the structure has the greater part of the α′-phase; it is thinner, which leads to increased values of hardness (350–360 HV) than in the central part of the columns (230–260 HV).

At the mesoscopic level, the specific texture of the material should be highlighted; it is due to the peculiarities of the thermal conditions at different melting points of the molten wire material during the plate growth. The typical columnar structure in the direction of growth of the specimen and the polygonal crystal structure in the scanning plane provide the observed increase in the strength properties of the formed material in comparison with the traditional analogous titanium alloys, which have less plasticity (Figure 14).

Figure 15 shows the strain curves, taking into account the changes in the cross-sectional area of flat specimens under uniaxial tension at the coordinates “σ_true_−ε_true_”. Analyzing the results shown in the figure, it can be seen that all the specimens cut from the upper and middle parts of the ingot (series A) along the direction of compression have similar mechanical properties, except for the specimen that was cut from the lower part of the ingot. This difference in mechanical properties is due to the different structural states of the lower sample, which was taken from the area close to the substrate.

These data show significant variations in the mechanical properties of specimens cut from the plate perpendicular to the printing direction (Series B). Specimens cut from the area near the edges of the plate have lower strength conversion values than those cut from the central area.

The ratio of strength to yield strength characterizes ductility. For the alloys studied, the ductility varies from 1.5 to 2. From the above data, it can be seen that the specimens of series A are more ductile. At the same time, the B-series specimens have a larger scattering of values σ_B_/σ_T_ and lower values than the A-series specimens.

The mechanical properties of the samples, when cut in different ways with respect to the compression direction, show a different combination of mechanical parameters reflecting the anisotropy of strength and plastic properties. In comparison with the data available in the literature (Table 8), the obtained samples have rather low mechanical properties. This shows that the Ti-6Al-4V alloy welding wire with reduced aluminum content (Ti-4Al-3V) is of limited use for obtaining products for industrial purposes. It can also be noted that the formation of a coarse-grained structure with low mechanical properties occurs when products are obtained by wire additive electron beam technology in general [13,31,32] (see Table 8), which, in this case, can be increased by conducting additional post-processing work. The reason for such a situation lies both in the grain size of the initial β-phase and in the structural phase state formed during cooling, along with their arrangement in the sample in the form of columnar elongated grains from the substrate to the top of the plate.

Figure 16 shows pictures (images 1–16) of the distribution of strain fields on the surface of specimens of series A and series B, taken in the process of the uniaxial tensioning of the specimen. On the basis of an analysis of the data presented in the figures regarding the distribution of deformation fields of relative transverse strain ε_XX_, longitudinal strain ε_YY,_ and shear strain ε_XY,_ it is possible to state that in the deformation patterns on the working surface of the specimens, the formation of randomly positioned local centers of tensile and compressive deformation (pictures 1ex–4ex, 1eu–4eu, and 1eu–4eu in Figure 16) is observed.

The transition from the elastic deformation domain to the initial stage of plastic deformation is reflected in the distribution of the deformation fields and is manifested by the appearance of extended areas of local deformation.

The patterns of the strain field distributions of relative transverse deformations in the specimens of series A, ε_XX,_ show the formation of extensive areas of compression deformation in the central part of the specimen (patterns 5ex–6ex, 5euu-6eu, and 5euu-6eu in Figure 16). In these areas, the local deformation of compression is modulo commensurable with the general averaged strain of tension over the whole working part of the specimen. The patterns of longitudinal strain distributions, ε_YY_, also show local areas of tensile strain at the same locations. On the shear strain fields, ε_XY_, which show strains in the central part of the specimen, completely different local strain regions are observed on the surface. These local strain centers are in the form of two long strain bands with different signs, which are perpendicular to the tensile axis (patterns 5echu–6echu in Figure 16). This arrangement of local deformation centers in the pattern of the strain field distributions in shear ε_XY_ deformations correlates with the arrangement of the microstructural layers formed as a result of plate production using additive technology (Figure 12, Figure 13 and Figure 14).

In Figure 16, in the patterns of the strain field distributions of relative transverse deformations ε_XX_ at the stage of a pre-break with an average longitudinal deformation <ε_YY_> ~10% in the whole working field of a specimen from the local center of plastic deformation in the central part of the specimen, a deformation of compression in a size comparable to average deformation <ε_YY_> (patterns 9xx–10xx) is observed. A similar situation is observed in the distribution of the strain fields of relative longitudinal strains, ε_YY_, at the stage of pre-breaking of the specimen; however, in the central part of the local center of plastic deformation, the tensile strain is almost 3–5 times larger than the average strain, <ε_YY_> (pictures 9eyy–10eyy).

In the central areas of the specimens, different distributions of the strain fields of relative shear ε_XY_ strains are observed at the point of specimen pre-break. In specimen 3 (series A, Figure 15a), the formation of two local shear foci of plastic deformation, extending perpendicular to the tensile axis and with different signs, is observed (Figure 16, pattern 10exy). Sample 1 (series A, Figure 15a) also shows the formation of two local shear foci of plastic deformation with different signs, but appearing in a more complex configuration (Figure 16, pattern 9exy). At this stage of deformation, the location of the local deformation foci in the pattern of strain field distributions of the shear ε_XY_ strains also correlates with the location of microstructure layers formed as a result of the electron beam surfacing of the wire during the additive manufacturing of the plate.

The study of the distribution of deformation fields of relative shear strains ε_XY_ at the pre-destruction stage on the specimens from series B, which are cut from the plate perpendicular to the printing direction, reveals the following features. It can be seen that in the lower gripping area, the localized foci of plastic deformation in the pattern of distribution of the deformation fields of relative shear strains ε_XX_ at the pre-destruction stage are extended along the vertical tensile axis (Figure 16 and photos 11 e_XX_–12 e_XX_).

The distribution of the strain fields of relative longitudinal strain ε_YY_ is also characterized by the localized foci of plastic deformation in the lower gripping area, where the strain is 4–5 times greater than the average longitudinal tensile strain over the entire working part of the specimen (Figure 16 and photos 11e_YY_–12e_YY_).

A comparison of the strain field distributions of the relative shear strains ε_XY_ at the pre-break stage of the specimens from the B series with specimens from the A series shows a significant difference in the plastic deformation process at the final stage. In the specimens from the B series, before fracture, the local deformation foci on the distribution of strain fields of relative shear strains ε_XY_ had a shape that was elongated in the direction parallel to the vertical tensile axis of the specimens. However, in the specimens from series A, the local deformation foci tend to extend in the direction perpendicular to the vertical extension axis of the specimens (Figure 16, image 9e_XY_–12e_XY_). The different orientations of local deformation foci in the patterns of the deformation field distribution in the shear ε_XY_ deformation specimens of series A and B reflect the complex nature of the plate macrostructure produced by the use of additive technology (Figure 12, Figure 13 and Figure 14).

The failure of the specimens occurred predominantly in the central part (Figure 17, Figure 18, Figure 19 and Figure 20) in different ways, depending on the orientation of the specimen. This was due to the structure of the specimens prior to testing, which were oriented differently in the fracture zone (Figure 17 and Figure 18). In addition to the differences in fracture structure, there was also a difference in the macroscopic shape change of the material in the fracture zone, which was more significant for the specimens tested in the vertical direction.

The results of the fractographic analysis of tensile fracture specimens show some differences in the characteristics of the fracture structure. In general, the picture in the fracture cross-section of the samples is represented by areas that are characteristic of ductile and partially quasi-brittle fracture, as evidenced by the presence of equiaxed pits and small scars or quasi-fractures.

Pitted ductile fractures are typical within the initial β-phase grains. At the boundaries of the α and β phase grains or plates, the formation of elements in the form of scars with the trace characteristic of a quasi-brittle fracture is possible. This is due to a high degree of deformation and hardening of the material during fracture.

## 4. Conclusions

The investigations that were carried out on the structure and properties of titanium alloy samples, obtained using wire additive electron beam technology with a welding wire with a composition of Ti-4Al-3V, show a large grain size, low defect content, and low strength. Large β-phase grains, elongated in the direction of heat dissipation, were formed in the specimens during pressing. During cooling, a mainly two-phase structure with a low β-phase content is formed. This situation leads to a decrease in the material’s mechanical properties, as observed in this paper. Plastic deformation and the fracturing of the specimens occurred unevenly throughout the tensile test. The distribution patterns of the various strain components showed columnar grains that were located longitudinally or transversely to the deformation axis. The test reveals high levels of relative local deformations in different areas, which significantly exceeded the total deformation of the specimen as a whole. This situation leads to the need to control the heat balance in the system during the pressing process, in order to reduce the grain size and change the structural phase state of the material. In the future, we intend to develop approaches to reduce the size of the formed grains, both by optimizing the energy input from the beam and the substrate cooling modes and also by revealing the influence of printing strategies, based on the sequences of beam movements in the scanning plane.

## Figures and Tables

**Figure 1 materials-16-03901-f001:**
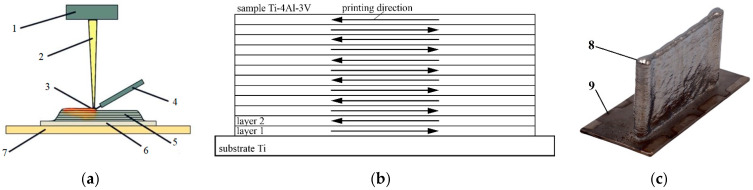
Scheme of wire cladding (**a**), print scheme (**b**), and general view of the printed plate (**c**). 1—Electron-beam gun; 2—electron beam; 3—melt bath; 4—wire feeder; 5—layer-by-layer deposited material; 6—substrate; 7—cooled table; 8—sample; 9—beam reversal zone.

**Figure 2 materials-16-03901-f002:**
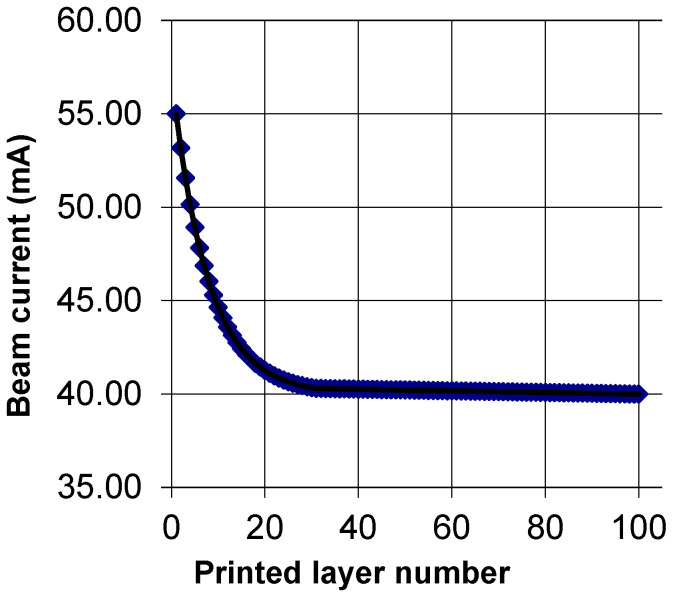
The change in electron beam current in the printing process, depending on the printed layer number.

**Figure 3 materials-16-03901-f003:**
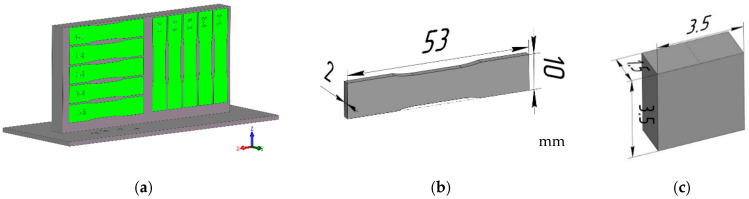
Scheme of the preparation of tensile samples from printed plates (**a**), characteristics of the tensile samples (**b**), dimensions of the micro-focus tomography samples (**c**).

**Figure 4 materials-16-03901-f004:**
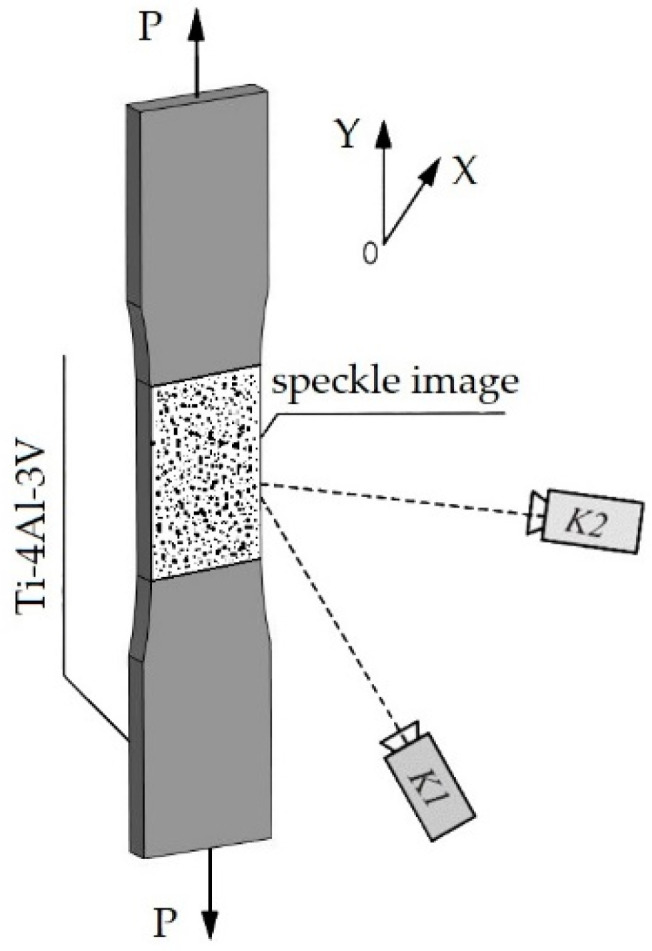
Schematic representation of the experiment during the tensile deformation of specimens with the recording of displacement fields using a speckled surface. K1 and K2—digital cameras; P—applied load.

**Figure 5 materials-16-03901-f005:**
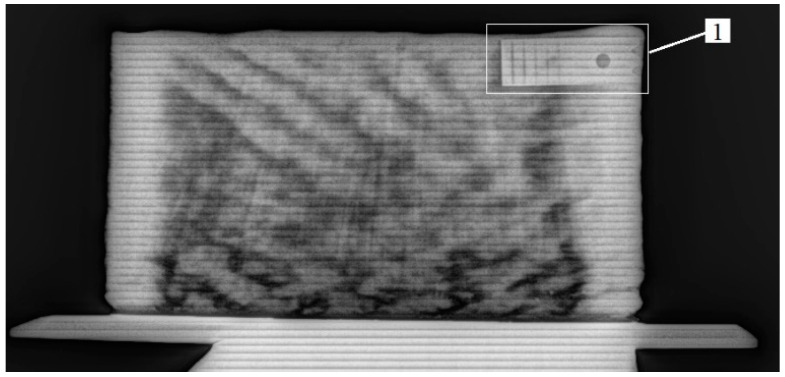
Radiographic image of the plate.

**Figure 6 materials-16-03901-f006:**
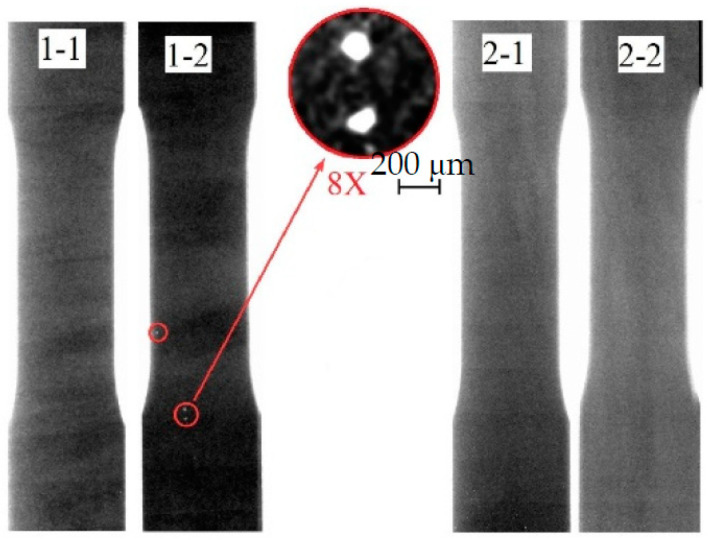
X-ray image of the tensile samples.

**Figure 7 materials-16-03901-f007:**
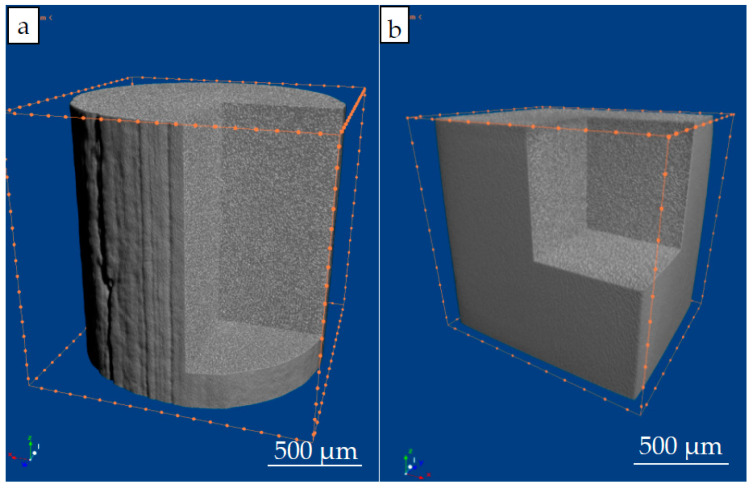
Tomographic image of the wire (**a**) and the printed sample (**b**).

**Figure 8 materials-16-03901-f008:**
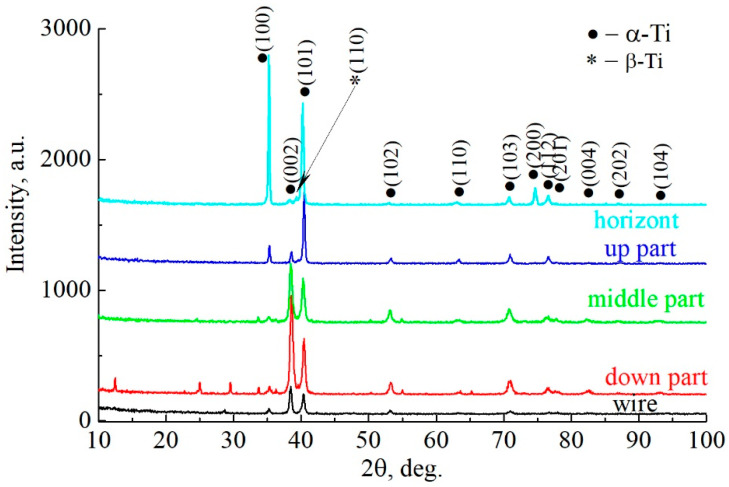
Comparison of the X-ray diffractograms obtained from the wire (black line) and different regions of the sample: down region (red line), middle region (green line), up region (blue line), and horizontally oriented region (turquoise line).

**Figure 9 materials-16-03901-f009:**
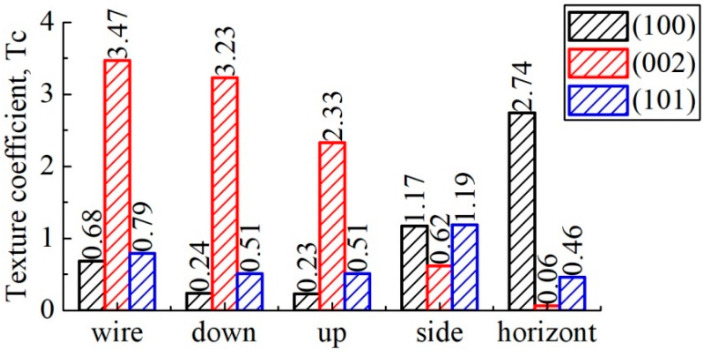
Change of texture coefficients.

**Figure 10 materials-16-03901-f010:**
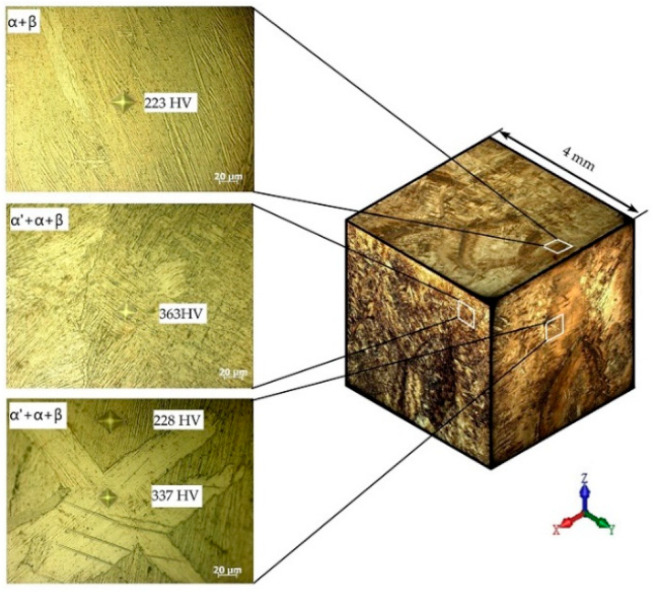
Alloy microstructure in three planes at low magnification (showing the sample center near the substrate).

**Figure 11 materials-16-03901-f011:**
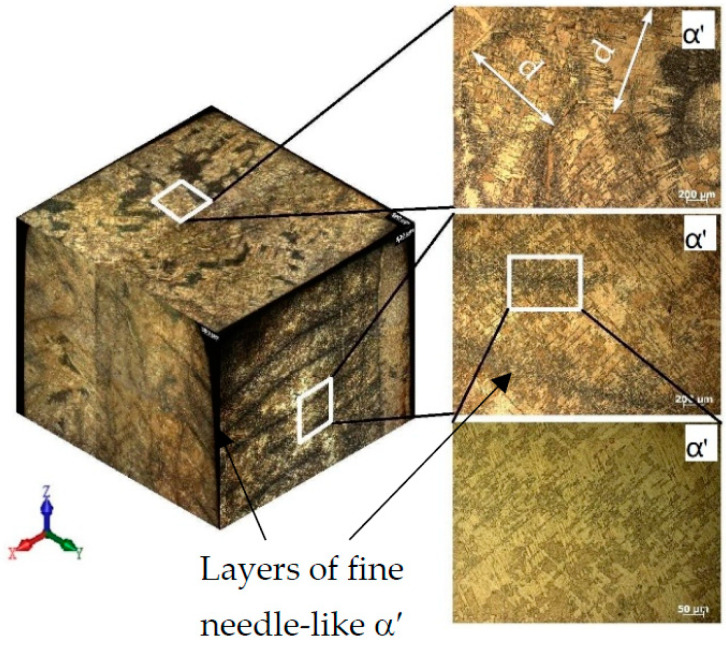
Alloy microstructure in three planes at low magnification (showing the edge of the sample near the substrate).

**Figure 12 materials-16-03901-f012:**
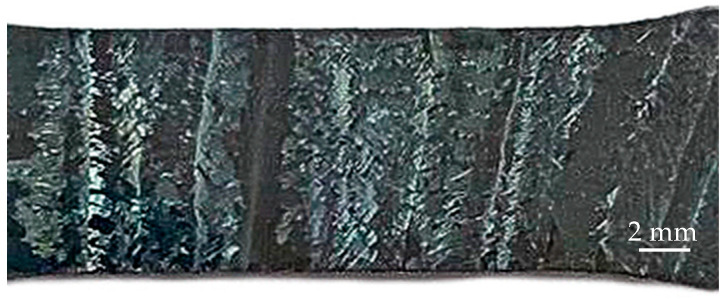
Image of the etched surface of the horizontal sample.

**Figure 13 materials-16-03901-f013:**
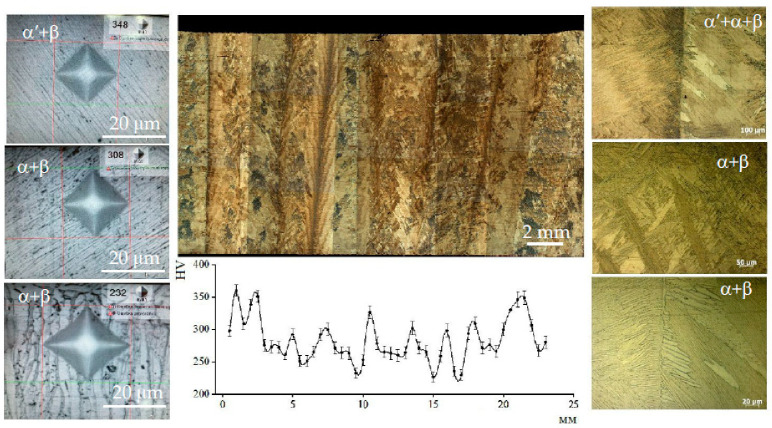
The columnar structure of a horizontal sample after etching and the microhardness measurement line, with steps of t = 0.5 mm.

**Figure 14 materials-16-03901-f014:**
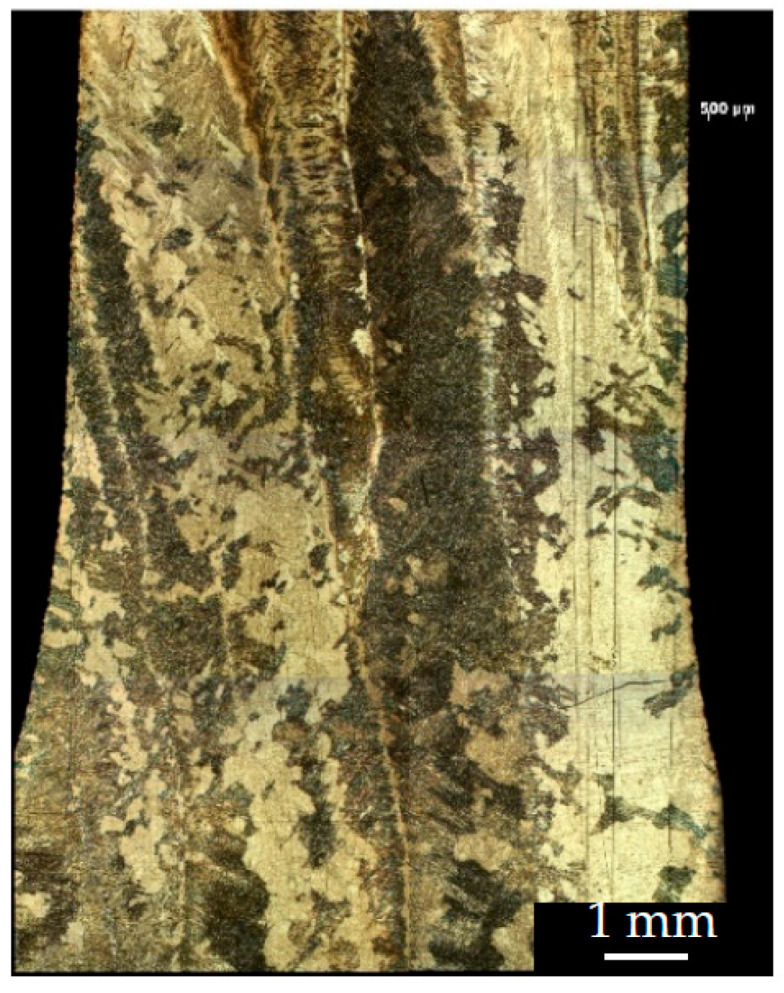
Image of the etched surface of the vertical sample.

**Figure 15 materials-16-03901-f015:**
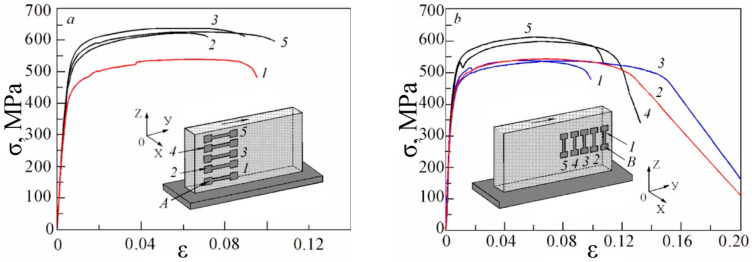
True strain curves under uniaxial tension of specimens in different directions ((**a**)—horizontal; (**b**)—vertical) relative to the printing direction, fused by additive technology.

**Figure 16 materials-16-03901-f016:**
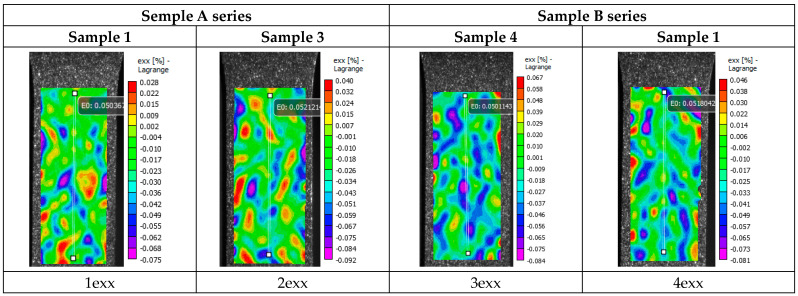
Distribution patterns of the transverse ε_XX_, longitudinal ε_YY,_ and shear relative strains ε_XY_ on specimen surfaces during the uniaxial tension of the specimens of series A and series B at different averaged strain values <ε_YY_> over the working field of the specimens: 1exx–4exx, 1eyy–4eyy, 1exy–4exy at <ε_YY_> = 0.05%; 5exx–6exx, 5eyy–6eyy, 5exy–6exy at <ε_YY_> = 1.0%; 7exx–8exx, 7eyy–8eyy, 7exy–8exy at <ε_YY_> = 0.97%; 9exx, 9eyy, 9exy at <ε_YY_> = 9.98%; 10exx, 10eyy, 10exy at <ε_YY_> = 9.34%; 11exx, 11eyy, 11exy at <ε_YY_> = 13.33%; 12exx, 12eyy, 12exy at <ε_YY_> = 10.14%. The sample numbers correspond to the curve numbers on the stress–strain relationships in Figure 15.

**Figure 17 materials-16-03901-f017:**
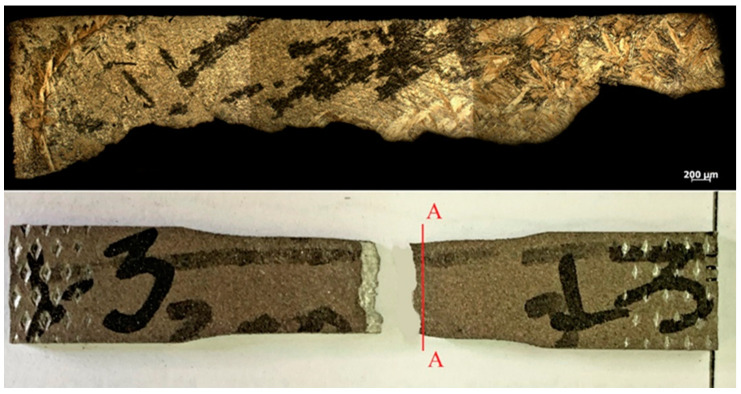
Fracture view of horizontal samples 1–4.

**Figure 18 materials-16-03901-f018:**
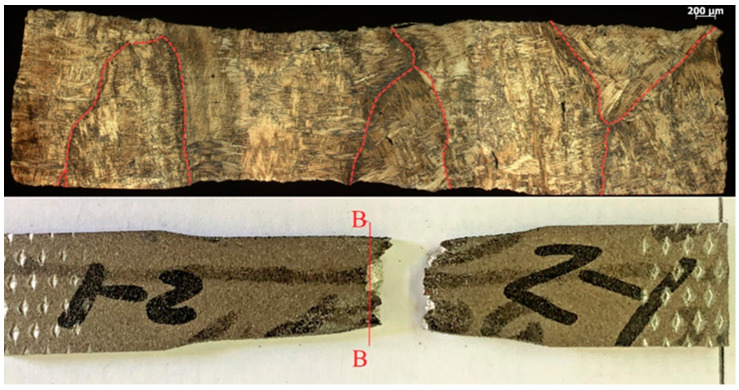
Fracture view of a specimen of vertical arrangements 2–3.

**Figure 19 materials-16-03901-f019:**
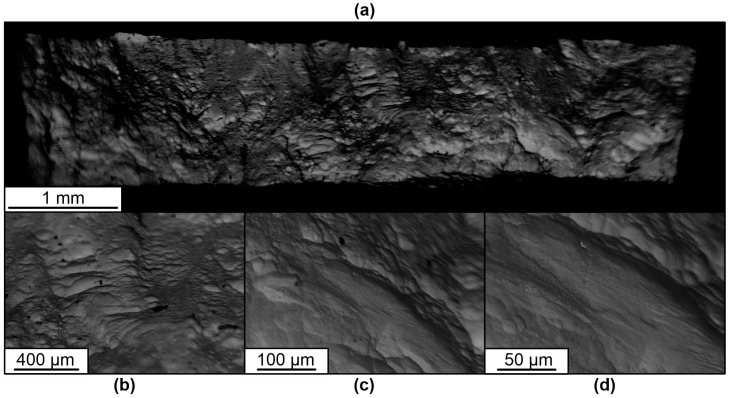
Fractography of the fracture surfaces of horizontal specimens under tension. (**a**)—Fracture view of the horizontal specimen 1–4 at low magnification; (**b**–**d**)—fracture view of the horizontal specimen 1–4 at high magnification.

**Figure 20 materials-16-03901-f020:**
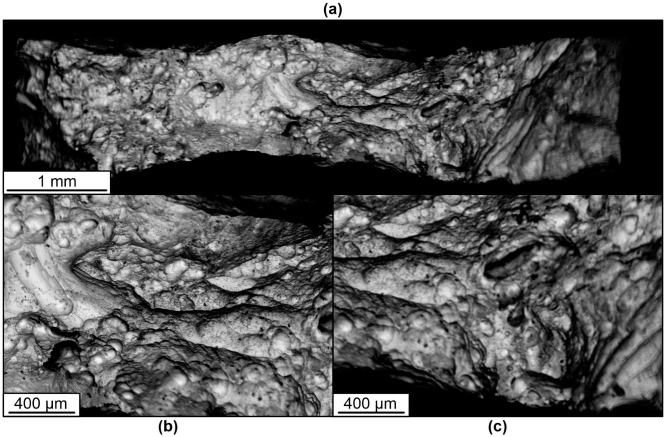
Fractography of the fracture surfaces of vertical specimens under tension. (**a**)—View of fracture of horizontal specimen 2–3 at low magnification; (**b**,**c**)—view of the fracture of horizontal specimen 2–3 at high magnification.

**Table 1 materials-16-03901-t001:** Chemical composition of titanium alloys (wt %).

Material		Ti	Al	V	Zr	Si	Fe	O	H	N	C	Other
Grade 2	Powder	base	–	–	–	0.10	0.25	0.20	0.010	0.04	0.07	0.10
	Cast	99.39					0.08					
Ti-6Al-4V *	Powder	base	5.3–6.8	3.5–5.3	0.30	0.10	0.60	0.20	0.015	0.05	0.10	0.30
	Wire											
Ti-4Al-3V *	Wire	base	3.5–4.5	2.5–3.5	–	0.10	0.15	0.12	0.003	0.04	0.50	0.30

* ISO 24034:2020.

**Table 2 materials-16-03901-t002:** The samples’ obtained parameters.

Parameter	Value
Substrate material	Grade2
Substrate dimensions, mm	150 × 60 × 2.5
Wire material	Ti-4Al-3V
Wire diameter, mm	1.6
Printed sample dimensions, mm	110 × 8 × 60
Layer thickness, mm	0.6
Layers amount	100
Accelerating voltage, kV	30
Beam current, mA	55–40
Beam sweep type	Circle
Beam sweep diameter, mm	3
Beam sweep frequency, Hz	100
Work table movement (printing speed), mm/min	400
Wire feed rate, mm/min	955

**Table 3 materials-16-03901-t003:** X-ray tube operating characteristics and controlled parameters.

Anode Tube Voltage, kV	90
Anode current, mA	4.5
Frame accumulation time, s	1
Basic IQI spatial resolution	W14
Control class according to ISO 17636	A
Resolution, μm	>200
Focus length, mm	250

**Table 4 materials-16-03901-t004:** Chemical composition of Ti-4Al-3V before and after 3D-printing.

Alloy	Ti	Al	V
Ti-4Al-3V (XRF, wire)	92.62 ± 0.58	3.88 ± 0.56	3.26 ± 0.18
Ti-4Al-3V (XRF material after printing)	92.88 ± 0.31	3.66 ± 0.28	3.13 ± 0.15

**Table 5 materials-16-03901-t005:** Calculated values obtained from the diffractogram of the wire sample.

Phase	Phase Amount, Vol. %	Lattice Parameters, Å, and Unit Cell Volume, Å^3^	CSR, nm	Microdistortions, Δd/d
α-Ti	92	a = 2.9605 ± 0.0030 c = 4.6577 ± 0.0090 c/a = 1.5733 ± 0.0200 V = 35.3535 ± 2.0	54 ± 26	1.9·10^−3^
β-Ti	2	-	-	-
Other phases	6	-	-	-

**Table 6 materials-16-03901-t006:** Calculated values obtained from the diffractogram of the sample, Ti-4Al-3V (lower).

Phase	Phase Amount, Vol. %	Lattice Parameters, Å and Unit Cell Volume, Å^3^	CSR, nm	Microdistortions, Δd/d
α-Ti	89	a = 2.9420 ± 0.0040 c = 4.6778 ± 0.0070 c/a = 1.5900 ± 0.0020 V = 35.0637 ± 3.0	29 ± 15	5.4·10^−3^
β-Ti	< 1	-	-	-
other phases	10	-	-	-

**Table 7 materials-16-03901-t007:** Calculated values obtained from the diffractogram of the sample Ti-4Al-3V (up).

Phase	Phase Amount, vol. %	Lattice Parameters, Å and Unit Cell Volume, Å^3^	CSR, nm	Microdistortions, Δd/d
α-Ti	94	a = 2.9427 ± 0.0090 c = 4.6889 ± 0.0090 c/a = 1.5934 ± 0.0150 V = 35.1636 ± 5.0	29 ± 20	6.6·10^−3^
β-Ti	<1	-	-	-
other phases	5			

**Table 8 materials-16-03901-t008:** Summary table of the mechanical properties of titanium alloys in the initial and printed states.

Material	State	Specifications			σ_B_/MPa	σ_0.2_/MPa	ε/%	Ref.
Grade 2	Rolled	Parallel to the rolling direction			408	277	38.7	[33]
Grade 2	Rolled	Perpendicular to the rolling direction			391	314	41.5	[33]
Grade 2	20 °C				544	350	-	[33]
Ti-6Al-4V	Cast	-			1000	896	8	[8]
	Rolled	-			1071	978	-	[9]
	SLM	As deposited		Horizontal	1006–1541	900–1326	2.7–12.3	[22]
				Vertical	1040–1380	664–1217	0.9–12.7	
		Annealed		Horizontal	1006–1066	906–981	7.5–13.6	
				Vertical	896–1116	772–1054	1.1–12.5	
		Stress relieved		Horizontal	1049–1086	852–981	6.5–13.1	
				Vertical	1032–1140	928–1070	2.7–10.5	
		HIPed		-	973–1035	883–942	6.3–19.4	
	EBM (DED)	Powder	As deposited	-	951.2	857.3	14.7	[21]
			750 °C	-	919	750	15.7	
			850 °C	-	917.2	703	24.8	
			950 °C	-	958.6	770	13.8	
		Wire	As deposited	Horizontal	907	847	11	[15]
				Vertical	846	800	14.5	
			Annealed (Vacuum, 650 °C, h., cooling in furnace)	Horizontal	916	858	11	
				Vertical	844	765	11.5	
			HIP (920 °C, 110 MPa, 2 h.)	Horizontal	809	729	13	
				Vertical	816	722	18	
Ti-Al-V-Cr-Mo-Zr	LPBF (DED)	As deposited			≈975	-	≈20	[24]
		480 °C (6h)			1611	-	5.4	
		520 °C (3h)			1452	-	7.8	
Ti-6Al-4V	EBM (DED)	Wire	Defect-free area	Horizontal	712	-	12	[13]
				Vertical	729	-	12	
			Area with defects	Horizontal	764	-	16	
				Vertical	792	-	16	
Ti-4Al-3V	EBM (DED)	Wire	As deposited	Horizontal	633	520	8.5	Current work
				Vertical	610	530	10.2	
Ti-4Al-3V	Drawing	Welding wire			665–865			GOST 27265-87, in Russian

## Data Availability

Data sharing is not applicable to this article.

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
