# Peer review of "Influence of the Coarse Grain Structure of a Titanium Alloy Ti-4Al-3V Formed by Wire-Feed Electron Beam Additive Manufacturing on Strain Inhomogeneities and Fracture"

_materials, 2023, doi:10.3390/ma16113901_

Round 1

Reviewer 1 Report

 In this manuscript, the authors investigated the effect of different phases of coarse grain structure on various properties of Ti-4Al-3V alloy fabricated by wire-feed electron beam additive manufacturing. The authors emphasize discussion about the anisotropic properties of the test specimens with supporting data. Regarding the quality and quantity of the results shown in this manuscript, several issues are required to be addressed before getting published.

Line 59-70:

The author uses too many “At the same time” in this section.

Line 84, Table 1:

Decimal numbers in this table (and also elsewhere in this MS) should be written with “ . ” instead of “ , “. For example, 0.10, 0.30, etc.

Figure 3:

The author should specify the unit of the numbers shown in the figure.

Line 133, Table 3:

The author should provide more description in the table title.

Line 153: The imaging range of 2θ angles was…

The author should use the word “scan range” instead of “imaging range” because XRD is not an imaging technique, but it utilizes a detector to sweep and collect the signal.

Line 174, Figure 4:

It is unclear if this figure was originally made for this MS or cropped from another source. This is because it contains a different font type and a fading background color with improper crop (some portion of the text and arrow are cut off).

Line 187:

What is “AT conditions”? The author should use full words when mentioning it for the first time in the MS.

Figure 5 and 6:

The author should include scale bar with the number and unit in both figures.

Line 200-202:

“The analysis of the elemental composition of the Ti-4Al-3V wire material showed a significant reduction of both Al and V in the titanium alloy, Table 4.”

The author should provide reasons for this statement. In fact, it is unclear that the differences of the numbers shown in table 4 are significant because there is no standard deviation included for each value.

Line 206:

“X-ray diffraction and X-ray fluorescence analyses…”

The author should provide information about XRF measurement in materials and method section.

Line 210 and 213:

Ti6Al4V should be written as Ti-6Al-4V.

Figure 8:

The author should offset diffraction pattern of wire away from x-axis line to make it easier to visualize. Some numbers representing crystallographic planes are overlapped.

Figure 10 and 11:

The author should use different text color on the images instead of red for better visualization.

Line 279-311:

From line 279-311, the author should address Figure 12-14 that correspond to the discussion in the context.

Line 315 and Figure 15 (in the figure caption):

Please correct non-English text.

Line 346-352:

The author should try to rewrite this paragraph with the addition of proper punctuation to make it easier to read.

Line 434-438:

The author should use symbols for alpha and beta words for better consistency in the context.

Author Response

Dear Mr./Ms. reviewer.

Thank you very much for you comments.

We agree with your comments and tried to correct them in the article.

Best regards,

Authors theam

Reviewer 2 Report

In this manuscript, the authors presented a wire-feed electron beam additive manufacturing technology applied to the fabrication of Ti-4Al-3V alloy. The microstructure, and mechanical properties were investigated as influence of the coarse grain structure. The authors showed many results, and some are interesting. However, the discussion is somewhat poor and could be improved, and there are some conclusions not supported by the results. Thus, I recommend this manuscript to be accepted for publication in Crystals after a minor revision. Some suggestions for the authors to improve the manuscript:

(1)   Line 304: In order to compare the grain size more intuitively, please keep the image multiples consistent.

(2)   Lines 285-286: To be consistent, add the change of samples in strength.

(3)   Lines 325-327: The discussion on the changes was not convincing, and the discussion could be improved.

(4)   Please check the correct standard of writing and format. Such as line 129 “(GE Sensing & Inspection Technologies, Germany)according to”, line 136 “for their micro- and macro-defects”, line 159 “10 ml HNO3”, line 161 “the Duramin-5''”.

(5)   Please keep the writing format consistent here, such as line 147 “(Fig. 3)”, line 179 “Figure 5”, line 186 “Fig. 6” et al.

(6)   Throughout the manuscript, the authors should check the word form “was/were” for the singular/plural cases.

 Please, minor editing of English language.

Author Response

Dear Mr./Ms. reviewer.

Thank you very much for you comments.

We agree with your comments, and tried to correct them in the article.

Best regards,

Authors theam

Reviewer 3 Report

·       Add the unit of measurement in Table 1, table 2

·       Add more recent references 2021-2022

·       In the Introduction section, the authors cited the specific results of previous research and cited them adequately. However, they did not mention their shortcomings in previous research. In the Introduction section, the penultimate paragraph should contain common features of previous research. The shortcomings of previous research should also be pointed out, in general.

·       In the Introduction section, the last paragraph should contain the scientific contribution and scientific hypotheses of your research. Complete, further elaborate the scientific contribution and scientific hypotheses of your research. Be explicit. In addition to the goal of the research (which was written), the novelty in the context of the scientific contribution should be pointed out. Scientific contributions should be written based on the shortcomings of previous research in the literature. In this way, the authors will better emphasize novelty and scientific soundness.

·       The introduction is very poor, you can mention a short comparation of the classes of biomaterials (advantages, disadvantages etc.). The following references are suggested: [1] https://doi.org/10.3390/bioengineering9110686; [2] DOI:10.3390/mi12121447

·       Discuss about mechanical properties of alloys 

·       In the conclusions, state the scientific contribution, the shortcomings of your methodology and future research.

Author Response

(The authors gave the same response as above.)

Round 2

Reviewer 3 Report

Paper was improved.